# Investigating the Effects of a Physical Education (PE) Kit Intervention on Female Adolescent Body Esteem

**DOI:** 10.3390/children10060938

**Published:** 2023-05-26

**Authors:** Sharon McIntosh-Dalmedo, Andrew M. Lane, Wendy Nicholls, Tracey J. Devonport

**Affiliations:** 1Sport and Physical Activity Research Centre, Faculty of Education and Health and Well-Being, University of Wolverhampton, Walsall WS1 3BD, UK; sharonymcintosh@yahoo.co.uk (S.M.-D.); t.devonport@wlv.ac.uk (T.J.D.); 2Centre for Psychological Research, Faculty of Education and Health and Well-Being, University of Wolverhampton, Wolverhampton WV1 1LY, UK; wendy.nicholls@wlv.ac.uk

**Keywords:** body image, appearance concerns, context, PE kit choice, exercise motives

## Abstract

It is widely acknowledged that adolescent females are particularly at risk of low body esteem. Low body esteem is associated with poor mental health and other negative outcomes. Interventions to help raise body esteem could have a considerable impact, especially if the intervention is low cost, easy to implement, and scalable. We investigated the efficacy of an intervention where participants could choose their clothes to wear during a physical education lesson (PE). PE is a context associated with low body esteem, a finding that is particularly evident among females. We hypothesized that body esteem would improve with choice. To show that body esteem does not randomly change, we tested its stability when assessed in a test–retest design when completed in a classroom setting, hypothesizing that body esteem would be stable. Participants (*n* = 110; Age M = 14.9 years; SD = 0.68 years) completed a 14-item body esteem scale eight times: (a) wearing the school uniform in a classroom and (b) during a PE lesson on two occasions in each context within a week. This was repeated at the re-test, which was separated by a two-week gap. The intervention was implemented and students were given a choice of PE kit and could wear their own (non-designer) clothes. The findings indicate that the choice of PE kit intervention was associated with improved body esteem in a PE context but was stable in a classroom context, which we hypothesized to be stable. We argue that this low-cost and scalable intervention represents a useful starting point for helping to support females with low body esteem among a potentially vulnerable population.

## 1. Introduction

Body esteem (BE) reflects the self-evaluations of one’s body or appearance [1]. Poor body image concerns remain a global phenomenon among adolescent females [2], which suggests that females are particularly at risk of experiencing low self-esteem. Body image dissatisfaction has been evidenced in half to three-quarters of adolescent females [3]. One suggestion to enhance positive body image is the provision of opportunities for the prevention of poor body image and intervention in schools [4]. It is widely acknowledged that poor body esteem during adolescence is correlated with a range of negative outcomes including poor mental health, low self-esteem [5], high anxiety [6], and poor quality of life [7]. Further, low body esteem during adolescence can continue into early adulthood [8]. Therefore, low body esteem among female adolescents is a worldwide issue, and identifying and implementing interventions to enhance body image is worthwhile [9,10,11,12].

Identifying ways to develop a healthy body image during the adolescent period for females remains a challenge [9,10,13,14]. Although evidence suggests that future research efforts should focus on enhancing body esteem in females, this is not necessarily the case. Body image is a rapidly growing line of investigation in males; however, the ideal models of male body image presented by media are very different from those of females, and antecedents and consequences of low body image are very different for females than males [10]. As attending school is compulsory for children in many countries worldwide, schools provide an environment where female adolescent body esteem attitudes are shaped [15]. Physical education, where students are often made to wear a PE kit to comply with school rules, and where children wear fewer clothes, is associated with low body esteem. Physical education can be a double-edged sword; on one hand, it is associated with positive effects and enhanced self-esteem through similar processes where physical activity enhances self-esteem [12,16], but on the other hand, it is associated with low body image. It is argued that the PE kit, that is, the official kit to be worn as part of school rules presents barriers to physical activity in adolescent females [17]. If wearing a PE kit is associated with low body esteem, this will have negative effects on attitudes toward doing physical activity, which is detrimental to long-term physical and mental health [16]. Arguably, opportunities for learning to enjoy physical activity are blighted as the PE environment is often a context that merely produces negative body image experiences [18].

Wang et al. [19] contend that situational or contextual influences are potent forces that influence attitudes toward one’s body. Cash [20] argued that there are multiple and interacting factors of behaviors that can be used to develop interventions [9,10]. Contextual change can invoke negative emotions, and body esteem can be impacted by numerous factors [21] including social scrutiny (e.g., other people evaluating your body), social comparisons (e.g., comparing your shape and size with others), body exposure (e.g., undressing in a PE changing room), being physically active (e.g., participating in school sport), and the type of clothing worn (e.g., PE kit) [20].

More recently, social media and ideals of body image combined with repeated social comparisons appear to have detrimental effects on body esteem [9]. In terms of interventions to support body esteem, evidence suggests that school-based interventions to improve girls’ participation have been inconsistent with varied effects [22,23,24]. To highlight the depth of the challenge, a meta-analysis of 62 body image interventions found only small-to-moderate improvements in female body image [25]. It has been argued that an issue with interventions is that they are not easily followed, possibly because they are too difficult to learn or carry out, or the participant is not sufficiently motivated to learn, and/or the participant does not have sufficient opportunities to practice the interventions [26]. Intervention conditions where participants are motivated, capable of implementing it, and have opportunities to practice could be scalable and, therefore, have considerable practical value against interventions that require expensive training or delivery [27].

The primary aim of this study was to assess the effects of a low-cost easy-to-implement intervention where participants had a choice of clothing to wear for PE as opposed to wearing the mandatory school PE kit. We compared body esteem scores assessed at baseline and post-intervention. We hypothesized that wearing a standard PE kit would reduce body esteem compared with wearing a school uniform. We also hypothesized that body esteem would be stable in the classroom/school uniform context. Finally, we hypothesized that the PE clothing intervention would be associated with improved body esteem. We suggest that within-subject analysis will show that body esteem is relatively stable within context and does not randomly change, but is not so stable that interventions could not be used to improve it. The design used in the present study controls for individual differences; therefore, if the intervention is effective, it provides a more rigorous contrast than a pre–post design, and also maintains ecological validity. By investigating the effectiveness of this simple intervention, we seek to provide evidence for an easy-to-use, cost-effective, and scalable intervention. The study set two delimitations, first, gender differences were controlled by selecting a female-only sample. Secondly, a single school was used, thereby providing a control for sub-cultures that could exist between schools.

A second aim was to re-examine the test–retest stability of the Body Esteem Scale [28]. Previous research conducted by Nevil et al. [29] demonstrated that subjective constructs assessed via self-report are prone to random error, and thus they encouraged researchers to check the relative stability of the scale being used. Body esteem, by nature, is expected to be sufficiently transient and change as a result of the effects of interventions, and scores on a self-report scale should reflect this. However, where the concept should remain relatively stable, such as when the context of completion remains the same, then scores on a self-report scale should confirm this.

## 2. Methods

### 2.1. Design

An experimental A-B-A within-subjects 2 × 2 design was used. The independent variables were time on two levels (baseline and post-intervention) and context on two levels (PE kit or uniform). All participants took part in the intervention which consisted of offering the students a choice of clothing to wear for PE. Dependent variables used to evaluate the effect of the intervention were the three subscales of the body esteem scale: BE-appearance, BE-weight, and BE-attribution [28]. The decision was not to try to offer an intervention in the uniform condition where there was no reason or active agent to suggest that body esteem would change. Therefore, we could contrast the effects of introducing an intervention in the PE condition where we hypothesized it would improve body esteem against the uniform condition, where we expected body esteem to be stable.

### 2.2. Participants

Volunteer participants were adolescent female school pupils (*n* = 110; age: mean = 14.9 years; SD = 0.68 years) from a school in the Midlands (U.K.). The first author is a teacher at the school and, therefore, discussions in terms of permissions from the senior management team in the school were eased; that is, there was trust between the researchers and the school to undertake a rigorous and ethically approved study. The school from which the sample was drawn is characterized by having a higher-than-average proportion of children falling within socioeconomic disadvantage (53% for the school versus 22.5% on average for all of the U.K. schools; DfE, 2022). The schools’ socioeconomic demographics outline that White British pupils represent approximately 62.2% of the school population and Black and Minority Ethnic populations represent approximately 37.8%.

### 2.3. Measure

To be able to capture adolescent female perceptions and experiences relating to body esteem, the 14-item Body Esteem Scale (BES; [28]) was selected. It is an adapted version of the Body Esteem Scale for Adolescents and Adults [1]. It is a multifaceted measure with subscales that are conceptually divergent. The measure differentiates an individual’s body esteem judgements concerning “Appearance” (general feelings about appearance), “Weight” (weight satisfaction), and “Attribution” (evaluations attributed to others about one’s body and appearance), thus enabling the measurement of specific dimensions of body esteem, opposed to that of a single construct. Within the construct of the measure, a six-item appearance subscale captures overall satisfaction with appearance; an example item is “I wish I looked like someone else.” The four-item weight subscale denotes satisfaction with one’s weight, for example, “Weighing myself depresses me” and the four-item attribution subscale summarizes evaluations attributed to others about one’s body and appearance. For example, “Other people consider me good looking”.

The Likert-scale items range from 1 (never) to 5 (always). After reverse scoring the appropriate items, participants’ responses are averaged across items. Higher scores are indicative of a more positive weight esteem. The measure is reported to provide good reliability and internal validity with a three-factor solution: attribution (the evaluations of one’s own body and appearance attributed to others), weight (weight satisfaction), and appearance (an overall feeling about one’s appearance). The three subscales have adequate reliability (appearance: alpha = 0.76; attribution: alpha = 0.68; weight: alpha = 0.84) and this measure has been validated for use among 11–16-year-olds (*n* = 674; M = 13.33 years, SD = 2.1 years) [28]. An acknowledged limitation of the present study is that the internal structure of the scale was not investigated, arguably through confirmatory factor analysis.

### 2.4. Procedure

Ethical approval was obtained from the University of Wolverhampton ethics committee (reference number: 117792). Parental/carer consent was sought for participants recruited from a secondary school in the Midlands (U.K.). At baseline, the 14-item BES [28] was completed in two contexts (uniform in a classroom and PE kit in a PE changing room) on two occasions within one week. No time limit was provided. Participants first completed the measure in a classroom, where they wore the standard school uniform (school jumper, trousers, and shirt and tie). The PE context was in changing rooms wearing PE kit (PE top, shorts, tracksuit bottoms, or skorts). This procedure was repeated two weeks later for the purpose of calculating the test–retest statistics at baseline.

At the intervention stage, for a period of two weeks, participants could choose to wear (1) a plain (with no logo) base layer long sleeve top (black, white, navy blue, or grey only), under their PE top, (2) plain (with no logo) black full-length leggings (opposed to shorts or a skort), and (3) their school jumper on top of their PE top. School jumpers were part of the school uniform and not part of the school PE kit. Wearing a standard school jumper would provide a level of consistency, and thus prevent participants from wearing different sporting logos and brands, for which it might be suggested that more expensive, branded clothing might influence outcomes by facilitating increases in body esteem, as opposed to the actual clothing items. During their standard PE lessons, the lesson content remained the same and no changes were made to the curriculum, as the activity had no bearing on the intervention. The intervention focus was entirely on the opportunity to choose from three additional clothing options or choose to wear their usual PE clothing.

Post-intervention, the BES was administered again—both in the uniform context and in the PE context. Both administrations of the questionnaire took place within a one-week period to allow for the test–retest statistics to be calculated from the follow-up data. In total, the measure was repeated eight times. While all 110 participants were given a choice over PE clothing, 90 opted to wear an additional item of a base layer, leggings, or a school jumper, with 20 participants opting to wear the standard PE kit.

### 2.5. Data Analysis

The data were cleaned and checked for errors. Allowances were made for missing data within the boundary so that only one missing item per subscale was accepted; otherwise, the data were removed from the study. Missing values were within an acceptable range (2.4%) according to previously reported guidance [30]. All mean scores were calculated. Descriptive statistics were calculated for the three subscales across the two conditions (PE kit and school uniform). To replicate the stability of the BES over time and within context, the BES was administered twice at each time point and in each context and correlations and t-tests were used to assess the stability of the measure within context.

To assess the effect of an intervention on BES scores, baseline scores on the BES were compared with post-intervention follow-up scores using a multivariate analysis of variance (MANOVA). To assess whether the context in which BES is measured (PE kit in PE changing room or classroom in school uniform) has any influence on the observed effect of the intervention, measures were taken in both contexts at both time points and the context was entered as an IV within the same MANOVA.

## 3. Results

Baseline: body image is stable in test–retest conditions

The results of the investigation of the relative stability of the BES indicate that all test–retest correlations (*n* = 12) were significant (r = 0.96–0.99, *p* < 0.001), and none of the *t*-tests demonstrated a significant difference between the test–retest scores (see Table 1). Therefore, the results support the hypothesis that BES provides a reliable measure.

### 3.1. Intervention Effects

Table 1 presents the means at baseline and post-intervention across contexts and subscales and Figure 1, Figure 2, Figure 3 and Figure 4 depict differences graphically. The dependent variables were the three subscales of the body esteem scale, being BE-appearance (“feelings about one’s general appearance”), BE-weight (“feelings about one’s weight”), and BE-attribution (“evaluations attributed to others about one’s body and appearance”). The highest mean body esteem scores were found for post-intervention data when wearing PE kit across all three subscales (M = 14.46, SD = 0.55) as opposed to wearing the uniform (M = 13.44, SD = 2.65). This shows the overall highest mean (and therefore greatest) body esteem scores were identified within the PE clothing intervention group when compared across all conditions: time, contexts, and as combined subscales. All the subscales of body esteem differed between those wearing the PE kit and those in uniform.

Across all subscales, the lowest (and therefore poorest body esteem scores) were consistently reported as BE-attribution despite time (baseline or post-intervention) or context (PE kit or uniform). The poorest BE-attribution scores were reported during the baseline testing (M = 11.10, SD = 1.89) and retest phases (M = 11.09, SD = 1.95). Regardless of time or context, the greatest overall appearance concern for adolescent females was not their own perceptions of their appearance or their weight but their perceptions of how other people see them. Across the subscales, the greatest increases (and therefore largest changes) were seen within BE-appearance in the PE kit from baseline to the post-intervention retest (M = 15.92, SD = 2.95 to M = 17.93, SD = 3.06); BE-weight from baseline to the post-intervention retest (M = 11.43, SD = 2.05 to M = 12.98, SD = 2.19), and BE-attribution from baseline to the post-intervention retest (M = 11.09, SD = 1.95 to M = 12.40, and SD = 2.09) (see Table 1). Therefore, the results show that wearing PE clothing is a moderator of body esteem.

### 3.2. Multivariate Analysis

MANOVA tested the effect of time (pre–post intervention) and context (PE kit and school uniform) on the three subscales measuring body esteem. The assumption of normality was tested using the Shapiro–Wilk test (*p* > 0.05) and partial eta-squared effect sizes. Normality checks were carried out on the residuals. The assumption of sphericity was tested. Where variables and interactions violated the assumption of sphericity, a Huynh–Feldt correction was applied. This was true in all cases except for the subscale * condition interaction.

The results show a significant interaction for changes in body esteem by context (PE v classroom) and time (pre–post) (Wilks Lambda = 0.40, F (3, 107) = 54.236, *p* < 0.01, partial η2 = 0.60). To illustrate the significant interaction effect, we depicted the results in Figure 1 which show the positive effects of the intervention with an increase in body esteem, whereas body esteem remained stable in the classroom context. Emphasis should be placed on the size of the effect size because the interaction was large [31].

A significant interaction was found for all subscales (BE-appearance: F (1, 106) = 73.35, *p* < 0.01, partial η2 = 0.40, see Figure 2; BE-weight: F (1, 106) = 51.24, *p* < 0.01, partial η2 = 0.32, see Figure 3; and BE-attribution: F (1, 106) = 63.31, *p* < 0.01, partial η2 = 0.37, see Figure 4). As observed in Figure 2, Figure 3 and Figure 4, the effect was more pronounced for the weight and attribution subscales, where a decrease in scores was observed in the uniform context, whereas an increase was observed in the PE context.

The results revealed a significant main effect for context (PE v classroom) (Wilks Lambda = 0.05, F (3, 107) = 35.180, *p* < 0.01, partial η2 = 0.38). As Table 1 shows, the average of all scales across all time points shows a slightly higher score for the PE kit than for uniform; this finding was influenced by the effects of the intervention. Univariate analyses demonstrated a significant effect was observed on all three subscales (BE-appearance: F (1, 107) = 66.01, *p* < 0.01, partial η2 = 0.38; BE-weight: F (1, 107) = 34.99, *p* < 0.01, partial η2 = 0.24; and BE-attribution: F (1, 107) = 37.61, *p* < 0.01, partial η2 = 0.26).

The main effect of time was not significant (Wilks Lambda =.94, F (3, 107) = 2.172, *p* = 0.1, partial η2 = 0.06). In univariate analysis, this pattern was found for all subscales (BE-appearance: F (1, 107) = 5.96, *p* = 0.02, partial η2 = 0.05; BE-weight: F (1, 107) = 0.09, *p* = 0.077, partial η2 = 0; and BE-attribution: F (1, 107) = 0.07, *p* = 0.80, partial η2 = 0).

Taken together, these findings indicate that the intervention had a significant and positive effect on body esteem. The increase in body esteem was only detected where measures were taken in the PE context following an intervention, and by contrast, body-esteem scores were stable in the uniform condition where no intervention was implemented.

## 4. Discussion

The present study developed and tested a simple to implement and scalable intervention designed to raise body esteem in adolescent females in a PE context.

We hypothesized that wearing a PE kit would reduce body esteem compared with wearing a school uniform [17]. The results of the present study support this assumption. These results contrast with results from our comparison of test–retest differences of body esteem scores in the school uniform condition where contextual factors such as place and clothes worn were consistent and show that body esteem was stable. Previous research has found that although physical activity has the potential to improve well-being including body esteem, such effects are not consistently found [22]; further, wearing a PE kit can have negative effects [23,24]. We hypothesized that offering students a choice would reduce these negative effects. The results support this hypothesis and findings show that offering students a choice of what to wear for PE offsets the negative effects found when wearing the PE kit. Further, the effect size for findings was large. Taken together, the results indicate the PE intervention was effective, and supports the notion that body esteem is relatively stable but can change following a targeted intervention.

We argue that this finding has considerable practical value as it is an intervention that is easy to implement and does not require complex behavioral change from participants [26]. It has been argued that for interventions to work they need to offer participants the opportunity to try the intervention, and clearly, scheduled PE classes offer this opportunity [26]. Michie et al. [26] argue that participants need to be motivated to carry out the intervention, meaning that they should be active in using the treatment. The results showing the uptake of the option to wear their own kit indicate that the participants were motivated to engage with the intervention. Should participants not wish to engage, this could be indicated by abstaining from PE or wearing the standard PE kit; however, most participants wore their own clothes and so were active participants in the intervention. Participating in interventions can often involve following an education program where participants are taught a range of psychological skills [10]. While such interventions have been found to be effective, they do increase cognitive load [27]. Cognitive load relates to the effort it takes to learn the intervention, and a downside of effort is fatigue and cessation. Interventions requiring participants to actively engage with learning materials also test the extent to which participants are motivated to overcome barriers such as fatigue [10,27]. Inconsistent effects of intervention could be partially explained by a lack of engagement in the active learning parts of an intervention. A benefit of the present study is that it was easy for participants to follow the intervention. This simplicity represents a strength as it is easy to use and as such is scalable. Participants need to be competent and able to implement the behaviors required. In this instance, this requires participants to own clothes they are happy to wear in a PE class; therefore, this behavior is likely to be achievable by most [26]. This creates a simple to implement intervention. An observation of many interventions to enhance body esteem is that they are complex, require a great deal of activity on behalf of the participant, and can be expensive as they require a practitioner to teach and provide feedback [25]. Many interventions are simply not sustainable on a wider scale, and difficult to effectively re-create and follow up post-intervention [32]. Notable additional challenges are also evident for schools in the form of financial restraints, limited staffing resources, and a lack of opportunities for intervention [33,34]. Interventions which are complex are often difficult to implement [26] and, therefore, many good ideas, in theory, fail to work in practice.

In line with our findings, the PE changing room is recognized as a context that may prompt and heighten concerns related to physical appearance [35], where negative emotions and comparisons magnify negative body esteem [36]. Our study outcomes also support the contention that social comparisons increase feelings of inferiority and self-consciousness [37]. Our results show that body esteem was lower when wearing a standard school PE kit compared with wearing a school uniform; that is, body esteem deteriorates when wearing the PE kit. In such contexts, it can be suggested that body image issues can be triggered as self-perceptions are fragile and can be manipulated [38] as adolescents internalize body appearance ideals [39]. Such experiences are consequential, and wearing a PE kit has the potential to increase feelings of self-consciousness and lower body esteem, particularly if poorly fitted [35]. Tight-fitting clothing has the capacity to increase body image concerns [40], notably, clothing and body satisfaction are inexorably linked [41]. Further, an impractical, poorly designed PE kit can limit and restrict participation in physical activities [42]. Body-conscious females continue to report appearance concerns related to PE kits that reveal body shape [43]. Yet a one style fits all approach continues to desecrate opportunities for positive adolescent female PE experiences. For the adolescent female to feel more body confident when physically active, a positive body image (regardless of size or shape) is a driving factor in facilitating improvements in attitudes toward physical activity [44]. In support of this, adolescence provides an opportune time to intervene due to the most dramatic decreases in female body esteem (early and mid-adolescence) for both appearance and weight esteem [45]. Our study outcomes imply that small but significant steps can be taken to empower girls through PE kit choice which increases comfort during lessons [43] and enhances positivity [46].

A few basic manipulations to the PE kit would strengthen and reduce negative perceptions and experiences within the context of physical education, i.e., reducing perceived physical differences in body size or shape [47]. Allowing the choice of PE kit may present protective mechanisms facilitating healthier positive body recognition and increased physical fitness in female adolescents [48]. Central to the intervention design is the ideology that when girls are motivated by choice and autonomy, sustained physical activity behaviors can increase [49].

Our findings indicate that clothing choice as an intervention can be influential. This offers a promising area of research; therefore, it could be hypothesized that improvements in PE clothing may lead to an enhanced positive experience within exercise and sport. We would therefore argue that what is needed is not highly sophisticated or innovative interventions; instead, rather simple, widely scalable, and effective interventions that will prove successful in engaging girls through greater satisfaction and experiences within physical activity are required [42,50].

In seeking ways to improve adolescent girls’ PE experience, previous assumptions were made that suggested girls’ negative attitudes were the overriding factor for their disengagement in PE activities, with limited investigations into why [51]. For the adolescent female, the benefits of being physically active are vast and associated with a wide range of positive attributes of body image [52,53]. Yet despite such benefits to body image, negative body image continues to be an obstacle to physical activity participation [54], as does engagement in girls’ PE lessons [55].

A second key finding from the present study is on the relative stability of the body esteem concept and the BES scales [1,28] in conditions where it is expected to be stable. The results of the present study support its test–retest stability. This is particularly important as results show that body esteem as a construct varies in intensity between different contexts. We suggest that the demonstration of test–retest stability adds considerably to the literature as it provides evidence that the scale can be used to test the effects of interventions and that changes over time are less likely to be due to measurement error [29].

Our study demonstrated that clothing has a significant effect on adolescent female body esteem. In comparison, being able to choose the PE kit had the effect of improving female body esteem opposed to wearing the standard school PE kit. Reporting body esteem when wearing the school uniform was relatively stable. Contextual differences may have been evidenced through changes to both the environment and clothing, which facilitated increases in negative body esteem. If we were to implement an intervention in the PE context, but only take the measures in the classroom, then no effect would have been observed.

### 4.1. Strengths of This Research

Having captured adolescent girl’s experiences of PE clothing and investigated the implications of contextual changes, these are promising findings. The experience of wearing PE clothing should not be detrimental to an adolescent females’ body esteem. Therefore, we can present an effective intervention that could be implemented across schools nationally and indeed internationally. In striving to meet this challenge, it was necessary to consider the challenges that school practitioners face in order to propose an intervention that could be presented as plausible. As such, this study may have far-reaching consequences for body esteem research in schools.

This intervention provides preliminary evidence in a limited research area. Through the suggested clothing manipulations, our body esteem intervention can be integrated with confidence into all PE programs. This presents minimum disruption to the PE curriculum, does not require specialist training or intervention specialists, can be administered whether facilities are limited or not, can be actioned regardless of constraints on a school timetable, can be implemented regardless of school financial budgets, and allows for stakeholders that include student voice, parents, and carers, teaching staff, senior leaders, and school governors to be part of the kit expansion options. This also recognizes the DFE, 2021 (Cost of School Uniform) statutory guidance which requires schools to make uniforms more affordable, and most importantly can be presented as an intervention that is inclusive of not only specific populations or year groups, but for every female adolescent in the U.K. (and beyond).

### 4.2. Limitations and Suggestions for Further Research

Although this intervention study is unique in its approach to identifying how female body esteem can be influenced through PE clothing, there are limitations that must be accounted for when interpreting these results. Firstly, further research is needed to investigate the impact of body esteem interventions across younger adolescent female ages, as study participants were Key Stage 4 in the U.K. pupils only (years 10 and 11). Secondly, data collection was based on a self-reporting measure, and qualitative research might provide a more extensive and in-depth analysis within this field. Thirdly, comparisons were conducted through repeated measures and not with a control group wearing exact clothing, and this may be seen as a limitation. However, this intervention was not exclusively about clothing and context; it was about the freedom to choose.

As these outcomes were recorded over a brief period of time, longitudinal post-intervention follow-up over 3, 6, and 12 months is required to investigate the sustainability of increases in body esteem over longer periods of time.

## 5. Conclusions

The study outcomes captured adolescent girl’s appearance experiences of PE clothing and body esteem. This research makes a significant contribution toward exploring the impact of contextual changes on body esteem through adolescent female perceptions and provides a basic way for schools to reinforce healthy body esteem within the context of PE. As such, these exciting results have important ramifications for PE practitioners striving to positively improve female adolescent body esteem experiences, through albeit minor adaptations. Our findings not only illustrate how contextual differences can impact body esteem but evidence why extending the range of PE clothing items available to female adolescents can improve body esteem. In addition, these favorable changes can be implemented with speed and delivered at scale within schools across the country. Further, these adjustments can be implemented in a sustainable manner, which is a major strength of this intervention. Effective body esteem interventions are required to improve female adolescent physical education experiences, and indeed both the development and efficacy of this intervention are promising.

## Figures and Tables

**Figure 1 children-10-00938-f001:**
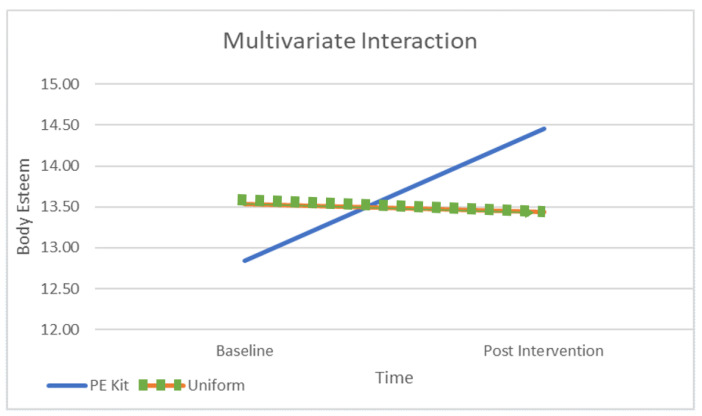
Multivariate Interaction of Context and Time on Body Esteem.

**Figure 2 children-10-00938-f002:**
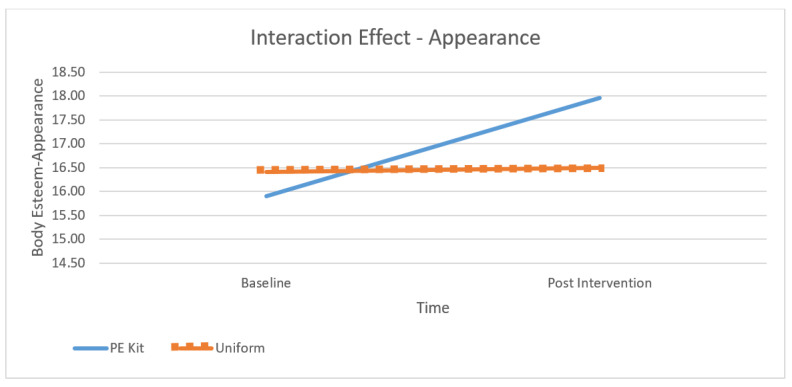
The effect of context and time on Appearance Subscale scores.

**Figure 3 children-10-00938-f003:**
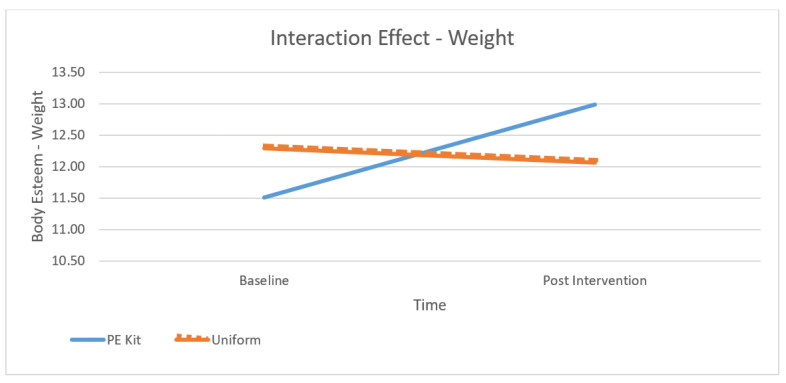
The effect of context and time on Weight Subscale scores.

**Figure 4 children-10-00938-f004:**
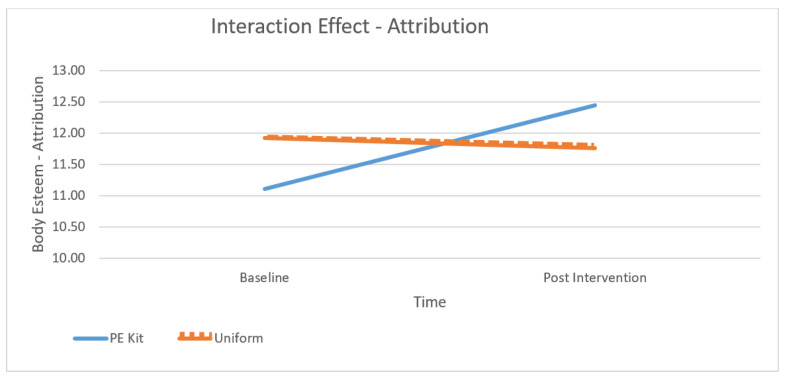
The effect of context and time on Attribution Subscale scores.

**Table 1 children-10-00938-t001:** Descriptive statistics and test–retest data for the Intervention Study (BE-appearance (feelings about one’s general appearance), BE-weight (feelings about one’s weight), and BE-attribution (evaluations attributed to others about one’s body and appearance).

			Test		Retest		Correlation Test–Retest				Difference Test–Retest	
Time	Context	Subscale	M	SD	M	SD	r	*p*	Confidence Interval		t	*p*
Baseline	PE Kit	Appearance	15.90	2.97	15.92	2.95	0.99	0.00	−0.11	0.08	−0.38	0.71
		Attribution	11.10	1.89	11.09	1.95	0.97	0.00	−0.08	0.10	0.21	0.84
		Weight	11.51	1.98	11.43	2.05	0.96	0.00	−0.02	0.19	1.53	0.13
		Total Mean	12.84	2.66								
	Uniform	Appearance	16.41	2.74	16.44	2.75	0.98	0.00	−0.12	0.07	−0.58	0.57
		Attribution	11.92	1.98	11.91	2.00	0.98	0.00	−0.06	0.08	0.26	0.80
		Weight	12.29	1.78	12.28	1.83	0.97	0.00	−0.08	0.10	0.20	0.84
		Total Mean	13.54	2.49								
Baseline		Total Mean	13.14	2.14	13.18	2.36						
Post-Intervention	PE Kit	Appearance	17.95	3.05	17.93	3.06	0.97	0.00	−0.11	0.16	0.40	0.69
		Attribution	12.45	2.02	12.40	2.09	0.97	0.00	−0.05	0.14	0.96	0.34
		Weight	12.98	2.18	12.89	2.19	0.96	0.00	−0.02	0.20	1.59	0.11
		Total Mean	14.46	0.55								
	Uniform	Appearance	16.49	2.62	16.51	2.42	0.97	0.00	−0.14	0.10	−0.29	0.77
		Attribution	11.75	2.07	11.72	2.03	0.97	0.00	−0.06	0.13	0.75	0.45
		Weight	12.06	1.84	11.98	1.77	0.96	0.00	−0.02	0.18	1.63	0.11
		Total Mean	13.44	2.65								
Post-Intervention		Total Mean	14.02	2.39	13.90	2.64						
	PE Kit	Total Mean	13.65	2.70	13.61	2.72						
	Uniform	Total Mean	13.49	2.30	13.47	0.38						

## Data Availability

The data presented in this study are available on request from the corresponding author. The data are not publicly available due to the focal population being children.

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
