# Peer review of "Investigating the Effects of a Physical Education (PE) Kit Intervention on Female Adolescent Body Esteem"

_children, 2023, doi:10.3390/children10060938_

Round 1
Reviewer 1 Report
Dear authors, some errors seen, or suggestions for improvement are shared:
Introduction
- It is necessary to include a higher percentage of previous literature from recent years, 2020, 2021 and 2022.
- There is clearly a lack of a scientific theory that supports this study, which is intended to complement-enrich or refute. In this case it would be a psychological theory (self-esteem), or psychoeducational.
- The general objectives are well established.
method
- 121-122. “and no control group was needed as participants acted as their own control”. This statement is not a justification. Having an equivalent control group will always give more rigor; but if it has not been taken into account, it must be justified why this method is used where the participants themselves act as control. All methods have advantages and disadvantages, so it is necessary to refer to them, without comparing with "others are not necessary...". This is contradictory to what was said in “Limitations and Suggestions for Further Research”.
- 127-135. Sampling, and not just the sample, should be explained and justified.
- 156-159. Better results have been found measuring reliability with McDonald's omega (instead of alfa's Cronbach).
McDonald's omega. Please see Hayes & Coutts (2020) and McDonald's (1999).
References:
Hayes, A.F., & Coutts, J.J. (2020). Use omega rather than Cronbach's alpha for estimating reliability. But…. Communication Methods and Measures, 14(1), 1–24.
McDonald, R.P. (1999). Test theory: A Unified Treatment. Mahwah, NJ: Lawrence Erlbaum. https://doi.org/10.4324/9781410601087
- 161. The code or reference of the document where the ethics committee approves the study must be entered.
- 201. Were the assumptions of normality and homogeneity of variances met to apply MANOVA? It must be specified.
- 203. There are two errors in the expression of (“M=”)
discussion
- Again, a discussion with/from a theory (of self-esteem?)
- Review more recent literature.
- Something should be discussed about the effect sizes found.
Congratulations are given to the authors for this study, which is considered relevant, and rigorous. But changes are necessary.
Author Response
Please see the attached file where we have addressed the reviewers comments. Please see also the revised article with track changes so that you can see the many changes made.
Reviewer one |
|
||
Introduction |
|
|
|
- It is necessary to include a higher percentage of previous literature from recent years, 2020, 2021 and 2022. |
We have updated the literature and provided more recent references where they are appropriate. |
|
|
- There is clearly a lack of a scientific theory that supports this study, which is intended to complement-enrich or refute. In this case it would be a psychological theory (self-esteem), or psychoeducational. |
We have outlined theoretical perspectives and findings. We are conscious not impose a theoretical framework on the study post-hoc. We have used Miche et al. work to frame why the intervention is useful in terms of it being easy to implement and scalable. |
|
|
- The general objectives are well established. |
Thank you |
|
|
method |
|
|
|
- 121-122. “and no control group was needed as participants acted as their own control”. This statement is not a justification. Having an equivalent control group will always give more rigor; but if it has not been taken into account, it must be justified why this method is used where the participants themselves act as control. All methods have advantages and disadvantages, so it is necessary to refer to them, without comparing with "others are not necessary...". This is contradictory to what was said in “Limitations and Suggestions for Further Research”. |
An experimental A-B-A within-subjects 2x2 design was used, and as such, there was no control group, as standard with a within subject design. We use the uniform condition as a contrast group, that is, no change was expected and by comparing with the intervention we will have more confident that findings do not occur by chance. |
|
|
- 127-135. Sampling, and not just the sample, should be explained and justified. |
The school was selected at the 1st author is a teacher at the school. We explained how this might data collection possible because there was trust between the researchers and the school. |
|
|
- 156-159. Better results have been found measuring reliability with McDonald's omega (instead of alfa's Cronbach). McDonald's omega. Please see Hayes & Coutts (2020) and McDonald's (1999). Hayes, A.F., & Coutts, J.J. (2020). Use omega rather than Cronbach's alpha for estimating reliability. But…. Communication Methods and Measures, 14(1), 1–24. McDonald, R.P. (1999). Test theory: A Unified Treatment. Mahwah, NJ: Lawrence Erlbaum. https://doi.org/10.4324/9781410601087 |
The Cronbach alpha statistics reported are from the referenced paper and not from the data set for the present study. |
|
|
- 161. The code or reference of the document where the ethics committee approves the study must be entered. |
We have provided the code/ |
|
|
- 201. Were the assumptions of normality and homogeneity of variances met to apply MANOVA? It must be specified. |
We have included the following in the method section. The assumption of normality was tested using the Shapiro-Wilk test (p > 0.05), and partial eta-squared effect sizes. Normality checks were carried out on the residuals. The assumption of Sphericity was tested. Where variables and interactions violated the assumption of Sphericity a Huynh-Feldt correction was applied. This was true in all cases with the exception of the subscale * condition interaction. |
|
|
- 203. There are two errors in the expression of (“M=”) |
We think these have been corrected. |
|
|
discussion |
|
|
|
- Again, a discussion with/from a theory (of self-esteem?) |
We have provided a greater amount of discussion |
|
|
- Review more recent literature. |
See 1 |
|
|
- Something should be discussed about the effect sizes found. |
Discussion |
|
|
Congratulations are given to the authors for this study, which is considered relevant, and rigorous. But changes are necessary. |
Thank you |
|

Reviewer 2 Report
Dear Authors,
I have read with great interest the manuscript received for review. Your research is valuable and well structured, the information presented is useful for researchers investigating adolescents. I can give you some minor suggestions to improve the evaluated version:
1. The citation style does not follow the requirements of MDPI journals, where references are cited by numbers in square brackets []. Citations in your manuscript are similar to APA style (where authors' names appear). I am attaching as an example an article recently published in the journal: https://www.mdpi.com/2227-9067/10/5/774
2. Line numbering is missing after table 1. Also, page numbering is wrong after table 1 all the way to the end of the document (page 8 is listed as page 1.................. ).
3. Charts 1-4 are not numbered or inserted correctly: Figures 1, 3, 2, 4.
4. It would be useful to specify the time frame in which you carried out the research (weeks/month/year).
5. The 14 items of the questionnaire could be included in an appendix (Weight/4, Appearance/6, Attribution/4). Readers would understand the details of your study more easily.
6. The study investigated only girls for the mentioned age range. A comparison of the results with a sample of boys would have been interesting....it is possible that gender generates differences for Body-esteem. I think that there are other independent variables that can significantly influence body esteem: involvement in physical activities (athletes vs non-athletes) or BMI classification (underweight vs normal weight vs overweight/obese). These aspects could be future research directions.
7. The data resulting from the application of MANOVA parametric techniques (Multivariate Analysis/page 8) would be easier to read if this information were summarized in tables (Wilks Lambda/λ, F Values, Sig., size effect/Partial Eta Squared -Ƞ2p). It's just a suggestion, their analysis in the manuscript is clear enough.
Author Response
|
I have read with great interest the manuscript received for review. Your research is valuable and well structured, the information presented is useful for researchers investigating adolescents. I can give you some minor suggestions to improve the evaluated version: |
|
|
1. The citation style does not follow the requirements of MDPI journals, where references are cited by numbers in square brackets []. Citations in your manuscript are similar to APA style (where authors' names appear). I am attaching as an example an article recently published in the journal: https://www.mdpi.com/2227-9067/10/5/774 |
References to be formatted |
|
2. Line numbering is missing after table 1. Also, page numbering is wrong after table 1 all the way to the end of the document (page 8 is listed as page 1.................. ). |
Line numbers have been added |
|
3. Charts 1-4 are not numbered or inserted correctly: Figures 1, 3, 2, 4. |
These have been reformatted |
|
4. It would be useful to specify the time frame in which you carried out the research (weeks/month/year). |
For a period of two weeks, the participants were given a choice of what to wear- this is included in the method (line 188-195). |
|
5. The 14 items of the questionnaire could be included in an appendix (Weight/4, Appearance/6, Attribution/4). Readers would understand the details of your study more easily. |
This is not a scale we have permission to reproduce, it is already available and we provide a reference. We have included some example items in the materials section (lines 168-173)
|
|
6. The study investigated only girls for the mentioned age range. A comparison of the results with a sample of boys would have been interesting....it is possible that gender generates differences for Body-esteem. I think that there are other independent variables that can significantly influence body esteem: involvement in physical activities (athletes vs non-athletes) or BMI classification (underweight vs normal weight vs overweight/obese). These aspects could be future research directions. |
Thank you we have noted this as a point in the discussion. |
|
7. The data resulting from the application of MANOVA parametric techniques (Multivariate Analysis/page 8) would be easier to read if this information were summarized in tables (Wilks Lambda/λ, F Values, Sig., size effect/Partial Eta Squared -Ƞ2p). It's just a suggestion, their analysis in the manuscript is clear enough. |
To avoid repetition/duplication, we’d rather not repeat the statistics in a table as they are already presented in narrative format. |

Reviewer 3 Report
The study deals with compulsory uniforms on school grounds and in physical education classes, and only physical education is mentioned in the title.
Some things need to be explained a little better. The compulsory school uniform is not common everywhere. Is it regulated by law by the school, city, or state? Why are males not part of the sample, this should be commented.
Line 122 -125
Historically, body image investigations have focussed on 122 young adults, often failing to capitalise on opportunities to investigate body image across 123 adolescent populations (Mellor et al., 2013). Therefore, one suggestion to enhance positive 124 body image remains with opportunities for prevention of poor body image and interven- 125 tion in schools (Fazel et al., 2014).
This text does not belong to the design of the study, but to the description of the sample.
The age data in the summary, table, and sample description are as follows:
(N = 110; Mage = 14.9; SDage = 0.68)
I think it would be clearer this way (N=110; age: mean=14.9, SD =0.68)
Line 131-134
The socio- economic demographic revealed a free school meals allocation of 53%. This is higher than the UK national average of 22.5% (DfE, 2022). The schools’ socio-economic demographics outline that White British pupils represent approximately 62.2% of the school population and BME populations approximately 37.8%.
Can only children from socially disadvantaged families receive free school meals, does this mean that there are more socially disadvantaged students in this school? What does the acronym BME mean (explain)?
Body weight is closely related to the assessment of one's own body image. Why were no data collected on the physical characteristics of the sample? Data on subjects' weight and body mass index would be very important for this type of study. Obesity is a major problem worldwide and is related to individual body perception.
Uniforming was introduced in schools to avoid class stratification. The problem was that some children could afford the best brand-name clothes or sports equipment, while poor children could not afford even the cheapest basic equipment. Eliminating mandatory school uniforms would help with body image assessments, but how would that affect the self-image of poorer students?
I think that this problem should be commented in this paper.
Author Response
|
Reviewer 3 |
|
|
The study deals with compulsory uniforms on school grounds and in physical education classes, and only physical education is mentioned in the title. |
We begin the article by mentioning that the issue is most noted in females. We have offered a stronger explanation as to why female only sample. |
|
Line 122 -125 |
Thank you – this has been moved. |
|
Historically, body image investigations have focussed on 122 young adults, often failing to capitalise on opportunities to investigate body image across 123 adolescent populations (Mellor et al., 2013). Therefore, one suggestion to enhance positive 124 body image remains with opportunities for prevention of poor body image and interven- 125 tion in schools (Fazel et al., 2014). |
This has been amended |
|
Line 131-134 The socio- economic demographic revealed a free school meals allocation of 53%. This is higher than the UK national average of 22.5% (DfE, 2022). The schools’ socio-economic demographics outline that White British pupils represent approximately 62.2% of the school population and BME populations approximately 37.8%. Can only children from socially disadvantaged families receive free school meals, does this mean that there are more socially disadvantaged students in this school? What does the acronym BME mean (explain)? |
Thank you for your point, which will ensure that the information provided is clearly interpretable from the international readership. Free school meals are means-tested in the UK for those over the age of 8, thus they provide an index of deprivation which can be used to compare the populations within schools in the UK. We have rephrased this statement such that the main point is clear, and interpretable as follows: The school from which the sample was drawn is characterised by having a higher-than-average proportion of children falling within socio-economic disadvantage (53% for the school versus 22.5% on average for all of the UK schools). BME is referring to Black Minority Ethnic groups and this has been reworded appropriately as follows. |
|
Body weight is closely related to the assessment of one's own body image. Why were no data collected on the physical characteristics of the sample? Data on subjects' weight and body mass index would be very important for this type of study. Obesity is a major problem worldwide and is related to individual body perception. |
We agree that obesity is a pressing public health concern, indeed by increasing body esteem, it is hoped that there will be fewer barriers to involvement in physical activity, however that was not an intended outcome of the present study. We agree that the body composition of the adolescents may contribute to their body esteem. We have used a within subjects design so that this does not confound our findings. We were reluctant to take biometric measures of adolescent females in a school setting, as it is sensitive and unlikely to be agreeable to the collaborating school. |
|
|
Thank you for your comment. The intervention, as described in the method, did not permit freedom entirely and stipulated that clothing should have no logo. Students were allowed to choose from a selection of clothing that the school had agreed upon. For example, plain black leggings, plain black shorts or plain black PE skirt. The focus here is on having a choice over what they wear – but that choice can be from a pool of pre-selected non branded items for the reasons identified by this reviewer.
|

Round 2
Reviewer 1 Report
The submitted manuscript has gained a lot in quality. However, three issues must be reviewed:
- The reliability coefficient of the data collection instruments used must be calculated. Cronbach's alpha cannot be that of the original scale from another study; It must be analyzed if the instrument has worked in this study with this sample, therefore the alpha must be calculated for the present sample. Also, it is suggested, as in the previous review, to use McDonald's omega.
- Figures 1, 2 and 3 must be presented with the full scale of the Y axis. If this is not done, the graphs are visually biased. The intersection crossing will look "flatter" or less pronounced when the entire Y axis is plotted (from the lowest possible value achievable by the participants, to the highest).
- 192-215. It must be specified how many items each questionnaire had.
After applying these questions, the article would be shareable with the scientific community.
Author Response
We thank the reviewers for their comments.
The reliability coefficient of the data collection instruments used must be calculated. Cronbach's alpha cannot be that of the original scale from another study; It must be analyzed if the instrument has worked in this study with this sample, therefore the alpha must be calculated for the present sample. Also, it is suggested, as in the previous review, to use McDonald's omega.
This BES was used across the four occasions (time on two levels (baseline and post-intervention) and context on two levels (PE kit or Uniform). We have acknowledged this as a possible limitation via:
‘An acknowledged limitation of the present is that the internal structure of the scale was not investigated, arguably through confirmatory factor analysis’.
We argued that we could have extended analysis to run confirmatory factor analysis holding equality constraints on relationships over time. However, as Nevill et al. (2001) highlighted that structural equation modelling can create overly complicated findings and run the risk of confusing the reader unnecessarily. Hence we veered away from such an approach.
Nevill AM, Lane AM, Kilgour LJ, Bowes N, Whyte GP. Stability of psychometric questionnaires. J Sports Sci. 2001 Apr;19(4):273-8. doi: 10.1080/026404101750158358. PMID: 11311025.
Whilst alternatives to using Cronbach's alpha to demonstrate reliability have been proposed, these statistical techniques are largely employed in scale development. McDonald's Omega is not employed as standard, and comparisons between McDonald's and Cronbach's alpha are not yet conclusive (e.g. see Malkowitz et al, (2023) https://www.sciencedirect.com/science/article/pii/S259029112200122X which states " Regarding the comparison of α, ωt, and GLB.fa, we would not conclude that this simulation study identified any one of the three as being superior to the others."
- Figures 1, 2 and 3 must be presented with the full scale of the Y axis. If this is not done, the graphs are visually biased. The intersection crossing will look "flatter" or less pronounced when the entire Y axis is plotted (from the lowest possible value achievable by the participants, to the highest).
We agree with this statement. We have revised the work by highlighting the effect size. The figures do illustrate the findings and importantly allow the reader to see an interaction, which we feel is why the figure is needed. We seek the reviewers guidance here because our option would be to remove the Figures.
- 192-215. It must be specified how many items each questionnaire had.
This BES identified in lines 192-215 is the same BES described in the measures section. This was consistently used across the four occasions (time on two levels (baseline and post-intervention) and context on two levels (PE kit or Uniform). The item specifications are identified in the methods section and highlighted below for ease of identification.